# Biofortification of Lettuce and Basil Seedlings to Produce Selenium Enriched Leafy Vegetables

Martina Puccinelli [1,*], Fernando Malorgio [1,2], Lucia Pintimalli [1], Irene Rosellini [3] and Beatrice Pezzarossa [3]

[1] Department of Agriculture, Food and Environment, University of Pisa, Via del Borghetto 80, 56124 Pisa, Italy
[2] Interdepartmental Research Center "Nutraceuticals and Food for Health", University of Pisa, Via del Borghetto 80, 56124 Pisa, Italy
[3] Research Institute on Terrestrial Ecosystems, National Research Council, Via G. Moruzzi 1, 56124 Pisa, Italy
[*] Correspondence: martina.puccinelli@agr.unipi.it

**Abstract:** Selenium (Se) biofortification of plants has been recognized as a good strategy to improve the nutritive value of vegetables and increase Se daily intake in humans. Identifying the most appropriate method to enrich plants is a key issue in the biofortification process. We tested a biofortification technique that produces Se enriched seedlings for transplant, yet barely modifies conventional cultivation techniques. Lettuce (*Lactuca sativa* L.) and sweet basil (*Ocimum basilicum* L.) were exposed to selenium by adding 0, 1 and 3 mg $L^{-1}$ (lettuce) and 0, 2 and 3 mg $L^{-1}$ (basil) of Se, as sodium selenate, to the growing substrate immediately after sowing. When seedlings reached an appropriate size, they were transplanted into the open field, and plants were grown until maturity. Lettuce and basil seedlings accumulated selenium without any reduction in leaf biomass at maturity. The highest dose of Se induced a higher antioxidant capacity and flavonoid content in both species at both sampling times. At maturity, biofortified plants still showed a higher leaf Se content compared to the control, and would be able to provide from 10% to 17% (lettuce) and from 9% to 12% (basil) of the adequate intake (AI) of Se.

**Keywords:** *Ocimum basilicum*; *Lactuca sativa*; sodium selenate; pre-transplant; substrate



## 1. Introduction

Biofortification of plants improves the nutritive value of vegetable products and decreases the deficiency of mineral elements in humans [1]. Selenium (Se) is an essential trace nutrient, and as a component of the amino acids selenocysteine and selenomethionine and a cofactor of glutathione peroxidase (GSH-Px; EC 1.11.1.9) [2], it is involved in several biological processes including thyroid hormone metabolism, antioxidant defence, and immune function [3]. The European Food Safety Authority (EFSA) recommend 70 µg $day^{-1}$ of Se as an adequate intake (AI) for adult men and women [4].

Selenium is a non-essential element in higher plants [5], but it can have beneficial effects [6,7]. Treatments with Se at low concentrations enhance oxidative stress responses, delay senescence, and improve yield in Se-enriched plants [8,9]. Selenium can counteract oxidative stress by improving the activity of antioxidant enzymes, such as glutathione peroxidase (GSH-Px) (EC 1.11.1.9) [10], and enhancing the synthesis of secondary metabolites, such as carotenoids, phenols [10], and vitamins [11–13]. At high concentrations, Se (VI) ($SeO_4^{2-}$) can interfere with sulphur metabolism, inducing toxicity, and a reduction in growth, whereas Se (IV) ($SeO_3^{2-}$) uptake was found to be reduced by phosphate [14]. Selenium can be actively taken up and transported by sulphate transporters [15] and replace sulphur in proteins, leading to the synthesis of selenocysteines and selenomethionines [16].

In general, selenium may have a positive effect on leaf quality, since it induces a lower nitrate [17,18] and a higher phenolic [19] and pigment content [20]. In basil plants, selenium treatments affect the quality of leaves, reducing the nitrate level [21], and increasing

rosmarinic acid [21], as well as the content of other phenols [13,21]. It also increases the concentration of essential oils [21] and antioxidant activity [13,22].

To identify the most appropriate method to enrich plants with selenium is a key issue in the biofortification process. The biofortification strategy should deliver the right amount of Se to the plants, and at the same time be environmentally safe. The main methods for the biofortification of crops are fertilizing the soil with selenium salts, soaking the seeds in a Se-enriched solution before planting, foliar or fruit spray, and hydroponic cultivation using a nutrient solution enriched in selenium. The use of selenium enriched fertilizers involves the addition of large amounts of selenium to the soil, thus making this methodology uneconomical and environmentally unsafe. Moreover, plant Se uptake from soil is affected by several factors which cannot be controlled, and which influence the efficiency of selenium absorption [23]. Although the addition of Se to the substrate in pre-transplanting is a very promising technique, it has only been proposed by Businelli et al. [23]. This methodology entails using a small supply of Se and could potentially produce biofortified vegetables and reduce the amount of the microelement released into the environment, with minimum modification of conventional cultivation techniques. We believe that this makes it a better option than soil fertilization [24], the enrichment of nutrient solution in hydroponics [22,25], or foliar applications [26].

Sweet basil (*Ocimum basilicum* L.) is a popular annual aromatic herbal plant used in medicine, to flavour food and to prepare sauces such as pesto [27]. Lettuce (*Lactuca sativa* L.) is widely cultivated for the fresh consumption of e whole heads or as baby-leaves [28]. Both species are widely cultivated in the open field or in the greenhouse.

Our main goal was to develop a biofortification technique that would produce selenium enriched seedlings for transplant with few or no modifications to conventional cultivation techniques. An experiment was conducted in which basil and lettuce were treated with Se immediately after sowing. When seedlings reached an appropriate size, they were transplanted in the open field, and plants were grown until maturity.

## 2. Materials and Methods

### 2.1. Plant Materials and Growing Conditions

Seeds of lettuce (*Lactuca sativa* L. cv. Pesciatina) and sweet basil (*Ocimum basilicum* L. cv. Geniale) were sown in trays filled with peat. The plant density of the trays was 480 plant $m^{-2}$, with 0.06 L of substrate for each plant. After 23 days from sowing, seedlings were moved to a greenhouse and grown under controlled environmental conditions at the University of Pisa, Italy. When the plants reached the height of 10–12 cm, which is suitable for the commercialization as young plants, they were hand-transplanted, and grown in open field (Bientina, Pisa, Italy) at a density of 20 plants $m^{-2}$ and grown until maturity. Drip irrigation was used to supply water three times a day for one minute. An organic fertilizer (Stallatico Bio Vigorplant: N 3%, organic N 3%, $P_2O_5$ 3%, organic C 25%) was applied to soil before transplant. Physical and chemical characteristics of the soil were determined by standard methods [29]. The main properties of the soil were as follows: clay 16.65%, silt 18.48%, sand 64.87%, pH 6.97, CEC 15.82 Cmol(+) $kg^{-1}$, EC 911 μS $cm^{-1}$, available $p$ < limit of quantification, available Ca 1.45 g $kg^{-1}$, available Mg 0.12 g $kg^{-1}$, available Na 0.16 g $kg^{-1}$, available K 0.24 g $kg^{-1}$, total C 0.12%, organic C 1.23%, N 1.2%. Cultivation events and the climatic parameters are shown in Table 1.

**Table 1.** Cultivation events and climatic parameters recorded during the experiment.

| Cultivation Event | Date | |
|---|---|---|
| Sowing date and selenate treatment | 21 May 2020 | |
| Transplanting/1st sampling | 13 June 2020 | |
| Maturity/2nd sampling | 3 July 2020 | |
| **Climatic Parameters** | **Seedlings Cultivation** | **Open Field Cultivation** |
| Mean air temperature (°C) | 27.0 | 21.6 |
| Mean air relative humidity (%) | 54.0% | 79.8% |
| Mean daily solar radiation (MJ m$^{-2}$ day$^{-1}$) | 15.9 | 23.0 |
| Cumulative solar radiation (MJ m$^{-2}$) | 381.1 | 438.2 |

### 2.2. Experimental Design

Treatment with Se was performed immediately after sowing by applying a solution containing 0, 1 and 3 mg Se L$^{-1}$ for lettuce, and 0, 2 and 3 mg Se L$^{-1}$ for basil, to the growing substrate. In the treatments with 1, 2 and 3 mg Se L$^{-1}$ the total amount of Se applied to each plant was 60, 120 and 180 µg, respectively. Selenium was added as sodium selenate and these concentrations were chosen based on the results obtained in previous biofortification experiments on lettuce [18] and basil [22,30].

For each species and treatment, three replicates were made, each one consisting of one tray, for a total of 36 trays. For open field cultivation of both plant species, 15 plants were transplanted for each treatment in 18 separated plots (three plots replicated for each treatment), each with five plants.

Two plant samplings were performed, the first at transplanting and the second at maturity. Lettuce and basil plants were cut just above the substrate (1st sampling) or soil (2nd sampling) level and analyzed for biomass production and for the content of selenium, micro- and macro elements, leaf photosynthetic pigments, flavonoids total phenols, and total antioxidant capacity. For each replicate, 80 and 5 plants were harvested at transplanting and at maturity, respectively.

### 2.3. Determinations

#### 2.3.1. Plant Growth

For each replicate, the fresh weight (FW) of lettuce and basil leaves was determined at transplanting and plant maturity, then samples were dried in a ventilated oven at 50 °C in order to avoid selenium volatilization losses, till constant weight and the dry weight (DW) was recorded.

#### 2.3.2. Selenium Content

The oven-dried ground samples were used for the determination of Se content according to the UNI EN13657:2004 [31] and UNI EN ISO 17294–2:2016 [32] methods for sample digestion and selenium determination, respectively. Three replicates were analyzed for each treatment by the CAIM group (Follonica, GR, Italy).

A mixture of hydrochloric acid and nitric acid was added to the sample in a ratio 10:1. The condenser was connected to a reaction vessel and the adsorption vessel was filled up with aqua regia. The mixture was warmed to the temperature of the reaction mixture and maintained for 2 h. After cooling of the vessels, the content of the adsorption vessel was added to the reaction vessel via the condenser, and a further 10 mL of diluted nitric acid was added. The content of Se in the digestion mixture was analyzed by Inductively Coupled Plasma Mass Spectrometry (ICP-MS).

#### 2.3.3. Mineral Content

The oven-dried ground samples were mineralized with a mixture (5:2) of nitric acid (65%) and perchloric acid (35%) at 240 °C in a microwave for 1 h and the mineral nutrient contents were analysed by inductively coupled plasma mass spectrometry ICP-MS (Na, K,

Ca, Mg, Cu, Fe, Mn, and Zn) and by UV/VIS spectrometry (P) [33]. A total of 100 mg of dry leaf samples were extracted in 20 mL of distilled water at room temperature for 2 h, and the extracts were used for the determination of the $NO_{3-}$ content using the salicylic-sulfuric acid method [34].

### 2.3.4. Leaf Photosynthetic Pigments, Flavonoids, Total Phenols, and Total Antioxidant Capacity

The content of chlorophylls, carotenoids, flavonoids, total phenols, and antioxidant capacity were analyzed in leaves of lettuce and basil plants at transplanting and at maturity. A total of 100 mg of foliar fresh tissues were extracted with 5 mL of 99% *v/v* methanol by sonication for 1 h and then maintained for 24 h at $-18\,^{\circ}$C. The methanol extract was used to determine spectrophotometrically the concentrations of total chlorophylls and carotenoids at 662.5, 652.4 and 470 nm. The sample concentrations were calculated by using the equation reported by Wellburn and Lichtenthaler [35].

To determine flavonoid content, 0.1 mL of the methanol extract was added to 0.06 mL of $NaNO_2$ (5%); 0.04 mL of $AlCl_3$ (10%); 0.4 mL of NaOH (1 M) and 0.2 mL of $H_2O$. Then, the absorbance was read at 510 nm. The results were expressed as mg catechin $g^{-1}$ FW [36].

The total phenol content was measured spectrophotometrically in the same methanol extracts using Folin–Ciocalteau reagent according to Kang and Saltveit [37]. The absorbance was read at 765 nm, and the concentration was calculated using a calibration curve containing 0, 50, 100, 150, and 250 mg gallic acid $L^{-1}$ and expressed as mg of gallic acid (GAE) $g^{-1}$ FW.

The ferric reducing ability of the plasma (FRAP) assay was used to determine antioxidant capacity, using an aliquot of methanol extract, according to Benzie and Strain [38]. The results were expressed as $\mu$mol Fe (II) $mg^{-1}$ FW.

### 2.4. Statistical Analysis Data

Data were tested for homogeneity of error variances with Levene's test, and subsequently were subjected to one-way ANOVA, with the Se treatment as the variable. Mean values were separated by Tukey's post hoc test ($p < 0.05$). Statistical analysis was performed using R Statistical Software, © The R Foundation, Vienna, Austria.

## 3. Results
### 3.1. Lettuce Plants
#### 3.1.1. Seedlings

The fresh (FW) and dry (DW) weight, and the dry matter content (DW/FW) of *Lactuca sativa* seedlings collected at transplanting were not negatively affected by treatments with Se (Figure 1A–C).

The addition of selenium to the substrate increased the Se concentration in leaves, and the highest value was obtained upon exposure to 3 mg Se $L^{-1}$ (+302% compared to treatment with 1 mg Se $L^{-1}$) (Figure 2A).

The treatments with Se led to an increase in the leaf content of Na (+19.1% and +26.7% at 2 and 3 mg $L^{-1}$ of Se, respectively, in comparison to the control) (Table 2), and Zn (+33.5% and +35.2% at 2 or and 3 mg $L^{-1}$ of Se, respectively, compared to the control) (Table 3). The highest dose of Se also increased the leaf concentration of Mg (+15.7%) (Table 2), Cu (+60%), Mn (+11.3%) and Fe (+27.2%) (Table 3), in comparison to the control. On the other hand, the leaf contents of $P-PO_4$, K and Ca were not influenced by the addition of Se (Table 2).

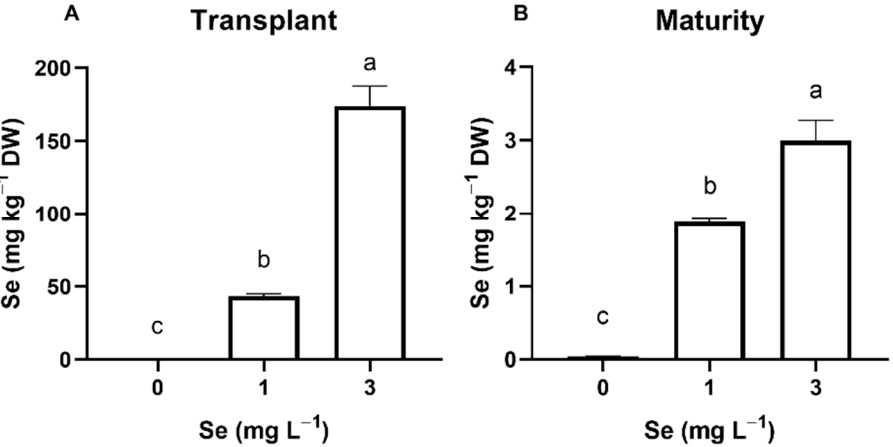

**Figure 1.** Fresh (FW) and dry (DW) weight, and dry matter percentage (DW/FW) of lettuce plants, measured at transplant (**A–C**) and maturity (**D–F**), treated with different concentrations of Se at pre-transplanting.

**Figure 2.** Selenium (Se) content in lettuce plants, measured at transplant (**A**) and maturity (**B**), treated with different concentrations of Se at pre-transplanting. Means (*n* = 3) flanked by different letter are statistically different for *p* = 0.05 after Tukey's test.

**Table 2.** Phosphate (P-PO$_4^{3-}$), potassium (K), sodium (Na), calcium (Ca) and magnesium (Mg) content, measured at transplant and at maturity, in lettuce plants treated with different concentrations of Se at pre-transplanting.

| | Se (mg L$^{-1}$) | P-PO$_4^{3-}$ (g kg$^{-1}$ DW) | K (g kg$^{-1}$ DW) | Na (g kg$^{-1}$ DW) | Ca (g kg$^{-1}$ DW) | Mg (g kg$^{-1}$ DW) |
|---|---|---|---|---|---|---|
| | 0 | 8.24 | 64.76 | 5.70 b | 8.15 | 3.75 b |
| | 1 | 6.76 | 59.90 | 6.79 a | 8.19 | 3.74 b |
| Transplant | 3 | 6.96 | 62.60 | 7.22 a | 8.37 | 4.34 a |
| | | | | ANOVA | | |
| | Se | ns | ns | * | ns | * |
| | 0 | 9.63 | 67.63 | 0.96 | 11.85 | 3.93 |
| | 1 | 10.43 | 66.19 | 0.80 | 10.25 | 3.25 |
| Maturity | 3 | 10.09 | 60.51 | 0.73 | 10.06 | 3.25 |
| | | | | ANOVA | | |
| | Se | ns | ns | ns | ns | ns |

Means (*n* = 3) flanked by the same letter are not statistically different for *p* = 0.05 after Tukey's test. Significance level: * *p* ≤ 0.05; ns = not significant.

The exposure of lettuce seedlings to 3 mg Se L$^{-1}$ led to a lower leaf total chlorophyll content (−16.6%) (Figure 3A), and a higher antioxidant capacity (+21.7%) and flavonoid content (+6.10%) (Figure 4A,C).

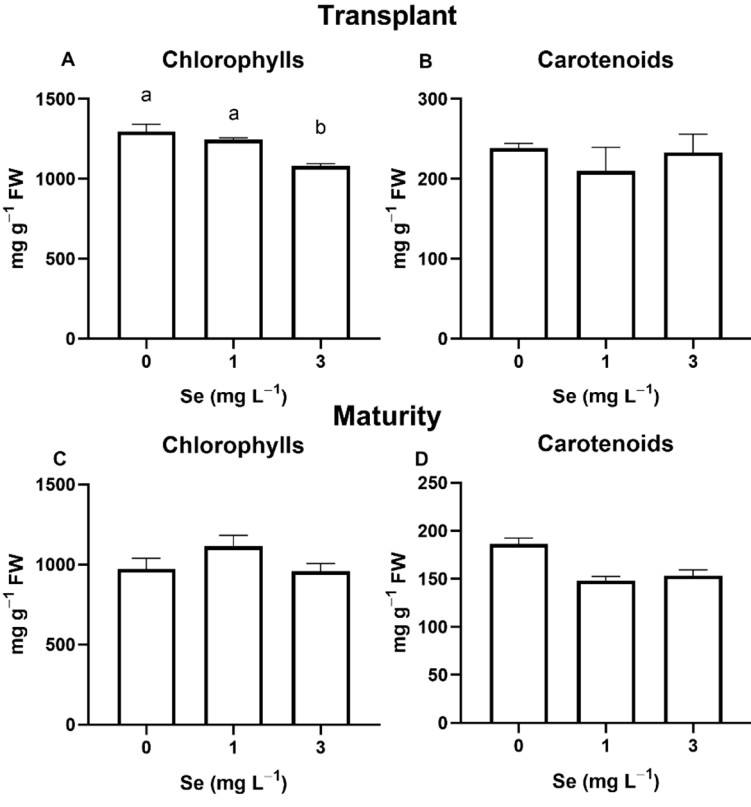

**Figure 3.** Total chlorophyll and carotenoid contents, measured at transplant (**A**,**B**) and maturity (**C**,**D**), in leaves of lettuce plants treated with different concentrations of Se at pre-transplanting. In subfigure (**A**), means (*n* = 3) flanked by different letter are statistically different for *p* = 0.05 after Tukey's test.

**Table 3.** Copper (Cu), manganese (Mn), iron (Fe) and zinc (Zn) content, measured at transplant and at maturity, in basil plants treated with different concentrations of Se at pre-transplanting.

| | Se (mg L$^{-1}$) | Cu (mg kg$^{-1}$ DW) | Mn (mg kg$^{-1}$ DW) | Fe (mg kg$^{-1}$ DW) | Zn (mg kg$^{-1}$ DW) |
|---|---|---|---|---|---|
| | 0 | 1.00 b | 82.0 b | 294.0 b | 119.3 b |
| | 1 | 1.33 ab | 88.0 ab | 248.0 b | 159.3 a |
| Transplant | 3 | 1.60 a | 91.3 a | 374.0 a | 161.3 a |
| | | ANOVA | | | |
| | Se | * | * | * | * |
| | 0 | 24.7 b | 72.7 b | 865.3 b | 72.7 b |
| | 1 | 30.7 a | 100.0 ab | 1147.3 a | 68.7 b |
| Maturity | 3 | 30.7 a | 127.3 a | 1162. a | 86.7 a |
| | | ANOVA | | | |
| | Se | ** | ** | * | * |

Means (*n* = 3) flanked by the same letter are not statistically different for *p* = 0.05 after Tukey's test. Significance level: ** $p \leq 0.01$; * $p \leq 0.05$; ns = not significant.

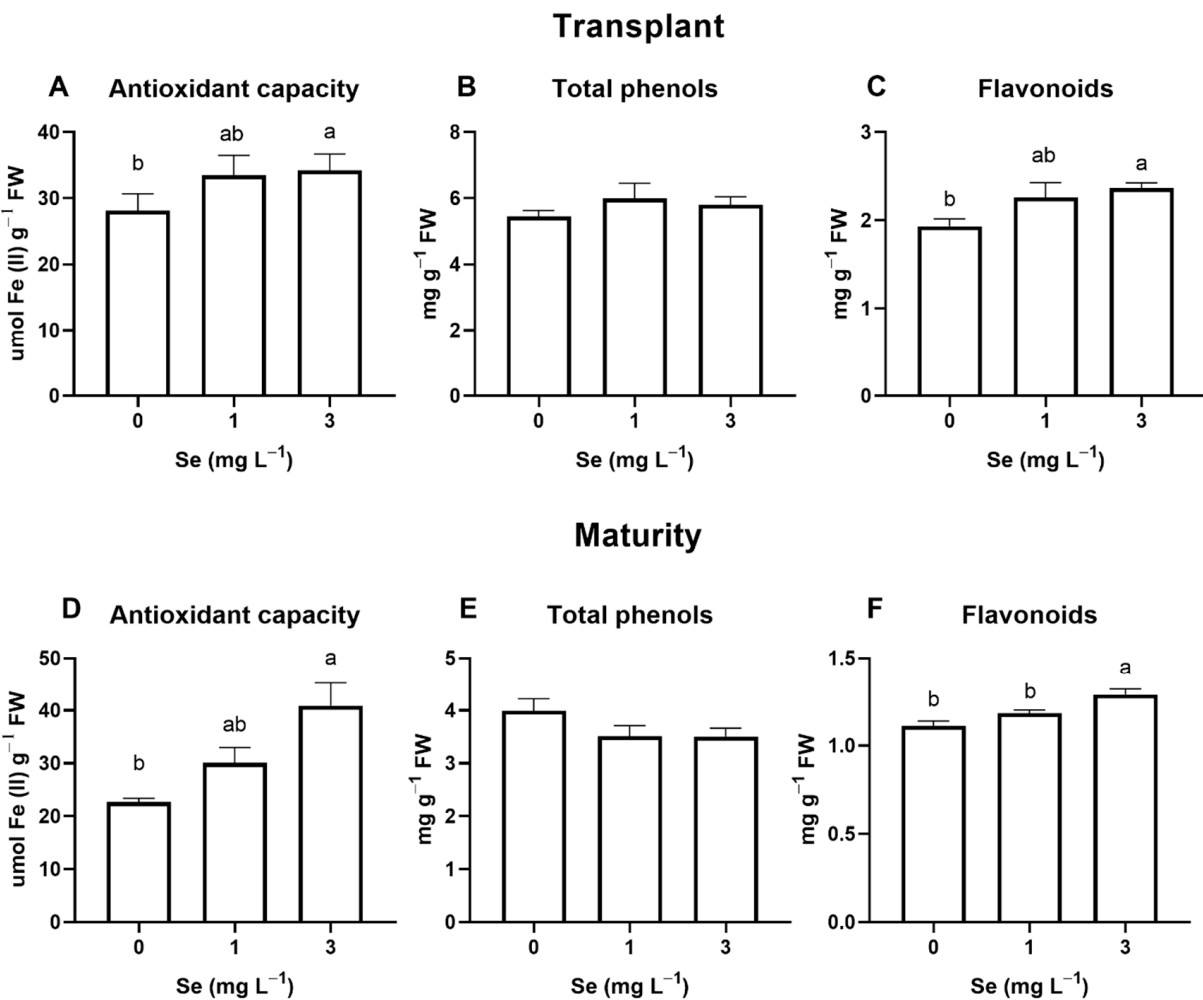

**Figure 4.** Antioxidant capacity, total phenol and flavonoid contents, measured at transplant (**A–C**) and maturity (**D–F**), in leaves of lettuce plants treated with different concentrations of Se at pre-transplanting. In subfigures (**A,C,D,F**), means (*n* = 3) flanked by different letter are statistically different for *p* = 0.05 after Tukey's test.

### 3.1.2. Mature Plants

The addition of Se to the substrate did not influence the fresh and dry biomass or the dry matter content of plants harvested at maturity (Figure 1D–F).

The Se concentration in leaves of Se-fortified plants generally tended to be lower in mature plants compared to seedlings. However, the total amount of Se accumulated per plant, calculated by multiplying the Se concentration by the plant dry weight, tended to increase at harvest compared to transplanting (Figure 2B).

The treatment with the highest dose of Se induced a lower leaf nitrate content (−20.0% compared to the control) (Figure 5), whereas the concentrations of P-PO$_4$, K, Na, Ca and Mg were not affected by exposure to Se (Table 2). Conversely, 2 and 3 mg Se L$^{-1}$ treatments increased the leaf content of microelements, particularly of Cu and Fe by 23.1% and 34.3%, respectively. In addition, the treatment with the highest dose of Se increased the leaf content of Mn (+75.1%) and Zn (+18.3%) (Table 3).

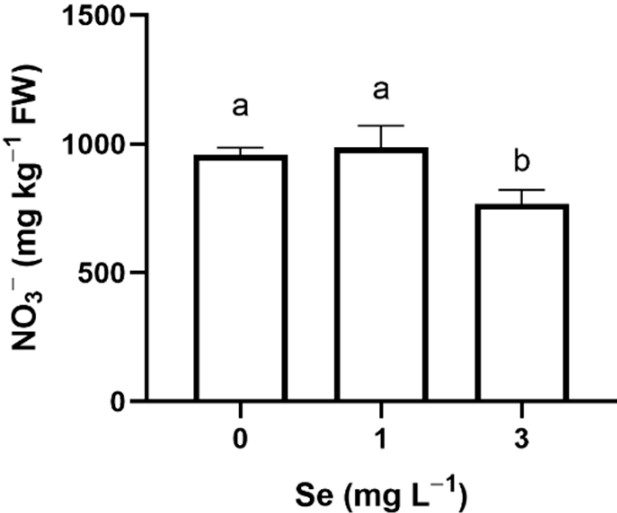

**Figure 5.** Nitrates (NO$_3^-$) content, measured at maturity, in leaves of lettuce plants treated with different concentrations of Se at pre-transplanting. Means (*n* = 3) flanked by different letter are statistically different for *p* = 0.05 after Tukey's test.

The contents of total chlorophylls, carotenoids (Figure 3C,D) and phenols (Figure 4E) in leaves of mature plants were not significantly different compared to the control, whereas plants treated with 3 mg Se L$^{-1}$ showed a higher antioxidant capacity (+81.0%) and flavonoid content (+15.2%) (Figure 4A,C).

### 3.2. Basil Plants

#### 3.2.1. Seedlings

The treatment with 3 mg Se L$^{-1}$ reduced the fresh weight of *Ocimum basilicum* seedlings, compared both to the control (−43.4%) and the 2 mg Se L$^{-1}$ (−40.4%) treatments, whereas it did not affect the production of dry biomass, thus resulting in an increase in dry matter content (Figure 6A–C).

The addition of increasing amounts of Se to the substrate resulted in a higher leaf Se concentration, with the highest value detected in seedlings treated with 3 mg Se L$^{-1}$ (+89.1%, in comparison to the 2 mg Se L$^{-1}$ treatment) (Figure 7A).

The seedling contents of K, Ca, Mg (Table 4), Cu, Mn, Fe and Zn (Table 5) were not affected by the addition of Se to the substrate, whereas both Se treatments induced a lower Na content, and only the exposure at 3 mg Se L$^{-1}$ reduced the concentration of P-PO$_4^3$ (Table 4).

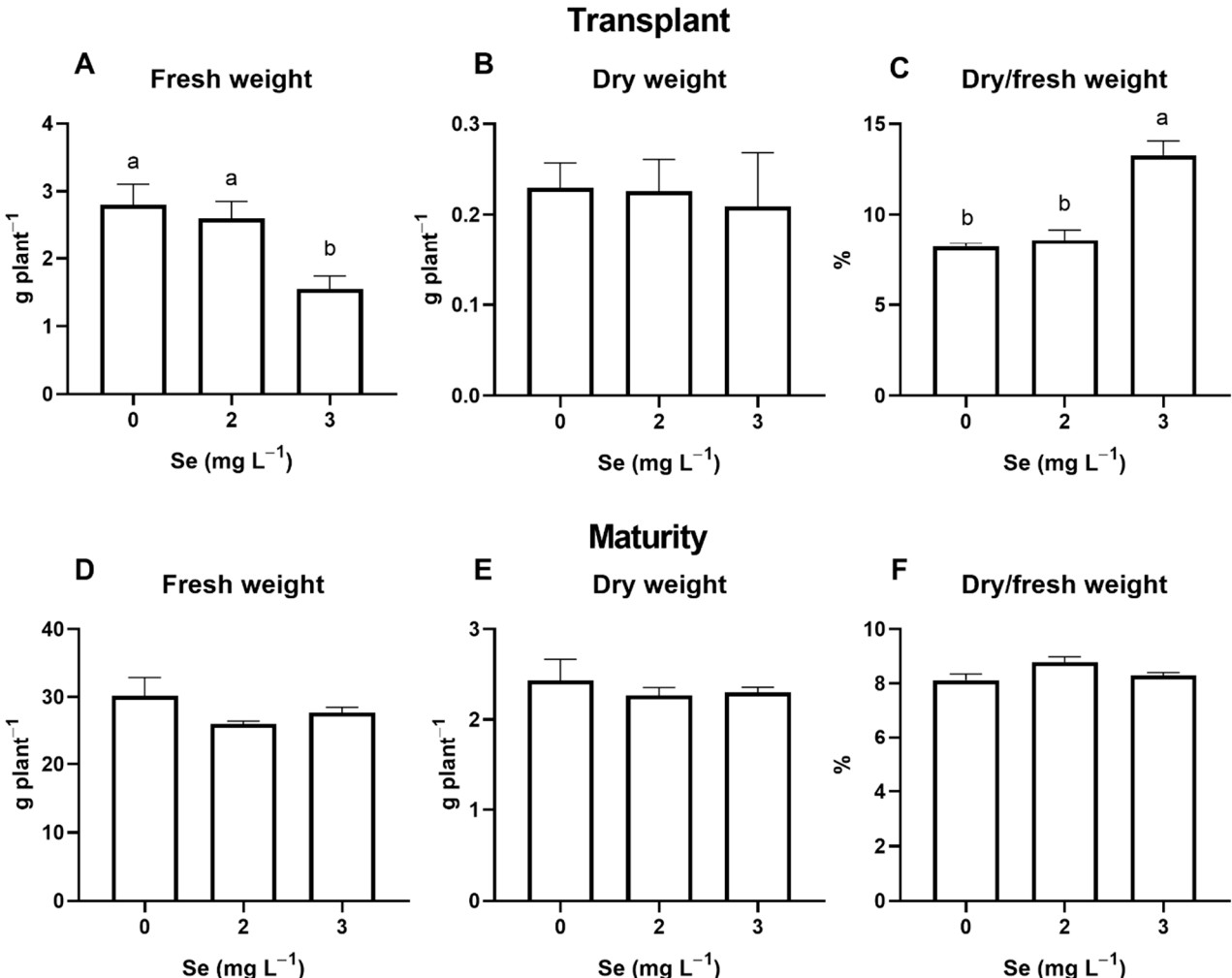

**Figure 6.** Fresh (FW) and dry (DW) weight, and dry matter percentage (DW/FW) of basil plants, measured at transplant (**A**–**C**) and maturity (**D**–**F**), treated with different concentrations of Se at pre-transplanting. In subfigure (**A**,**C**), means (*n* = 3) flanked by different letter are statistically different for *p* = 0.05 after Tukey's test.

**Table 4.** Phosphate (P-PO$_4^{3-}$), potassium (K), sodium (Na), calcium (Ca) and magnesium (Mg) content, measured at transplant and maturity, in basils plants treated with different concentrations of Se at pre-transplanting.

| | Se (mg L$^{-1}$) | P-PO$_4^{3-}$ (g kg$^{-1}$ DW) | K (g kg$^{-1}$ DW) | Na (g kg$^{-1}$ DW) | Ca (g kg$^{-1}$ DW) | Mg (g kg$^{-1}$ DW) |
|---|---|---|---|---|---|---|
| Transplant | 0 | 6.20 a | 44.97 | 1.30 a | 16.19 | 2.84 |
| | 1 | 5.84 a | 39.00 | 0.16 b | 13.91 | 2.54 |
| | 3 | 4.98 b | 40.43 | 0.23 b | 14.91 | 2.41 |
| | | | | ANOVA | | |
| | Se | * | ns | ** | ns | ns |
| Maturity | 0 | 5.80 | 48.17 | 0.02 | 23.43 | 4.56 b |
| | 1 | 3.96 | 41.50 | 0.01 | 24.41 | 4.87 ab |
| | 3 | 4.64 | 43.73 | 0.01 | 22.56 | 5.15 a |
| | | | | ANOVA | | |
| | Se | ns | ns | ns | ns | * |

Means (*n* = 3) flanked by the same letter are not statistically different for *p* = 0.05 after Tukey's test. Significance level: ** *p* ≤ 0.01; * *p* ≤ 0.05; ns = not significant.

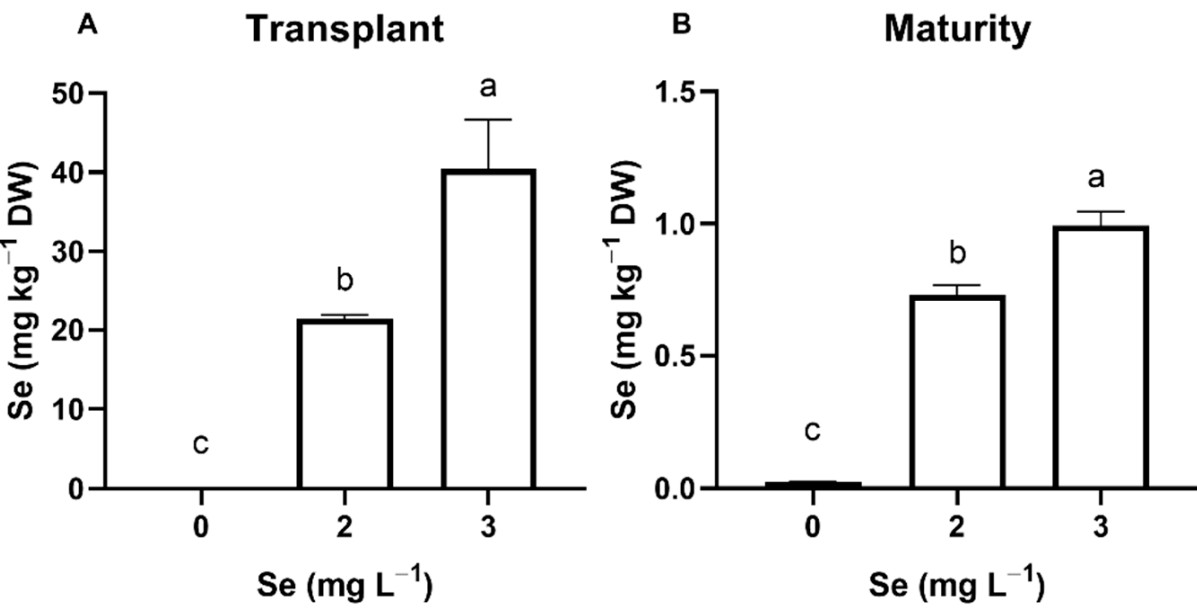

**Figure 7.** Selenium (Se) content of basil plants, measured at transplant (**A**) and maturity (**B**), treated with different concentrations of Se at pre-transplanting. Means (*n* = 3) flanked by different letter are statistically different for *p* = 0.05 after Tukey's test.

**Table 5.** Copper (Cu), manganese (Mn), iron (Fe) and zinc (Zn) content, measured at transplant and maturity, in basil plants treated with different concentrations of Se at pre-transplanting.

|  | Se (mg L$^{-1}$) | Cu (mg kg$^{-1}$ DW) | Mn (mg kg$^{-1}$ DW) | Fe (mg kg$^{-1}$ DW) | Zn (mg kg$^{-1}$ DW) |
|---|---|---|---|---|---|
| Transplant | 0 | 1.6 | 84.7 | 218.7 | 130.7 |
|  | 1 | 1.3 | 69.3 | 152.0 | 113.3 |
|  | 3 | 2.0 | 84.7 | 179.3 | 133.3 |
|  |  | ANOVA | | | |
|  | Se | ns | ns | ns | ns |
| Maturity | 0 | 15.3 a | 120.7 a | 449.3 a | 94.0 a |
|  | 1 | 8.0 b | 78.7 b | 372.7 ab | 80.7 b |
|  | 3 | 7.3 b | 82.7 b | 345.3 b | 82.7 b |
|  |  | ANOVA | | | |
|  | Se | ** | * | * | * |

Means (*n* = 3) flanked by the same letter are not statistically different for *p* = 0.05 after Tukey's test. Significance level: ** $p \leq 0.01$; * $p \leq 0.05$.

Both treatments with Se reduced total chlorophyll content, and exposure to 3 mg L$^{-1}$ decreased the total carotenoid content of seedling leaves (Figure 8A,B). However, a higher antioxidant capacity was detected in seedlings treated with 2 and 3 mg Se L$^{-1}$, and a higher flavonoid content when seedlings were exposed to 3 mg Se L$^{-1}$ (Figure 9A,C).

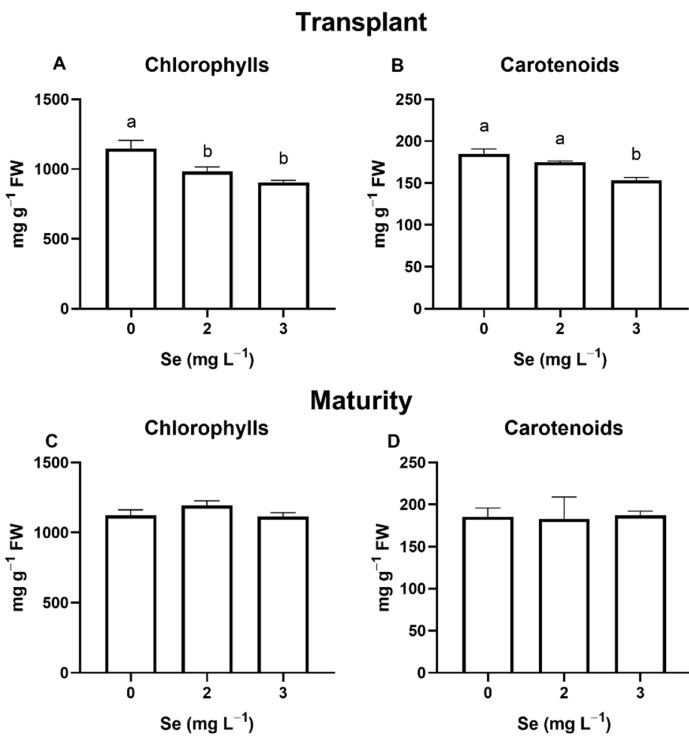

**Figure 8.** Total chlorophyll and carotenoid contents, measured at transplant (**A**,**B**) and maturity (**C**,**D**), in leaves of basil plants treated with different concentrations of Se at pre-transplanting. In subfigures (**A**,**B**), means (*n* = 3) flanked by different letter are statistically different for *p* = 0.05 after Tukey's test.

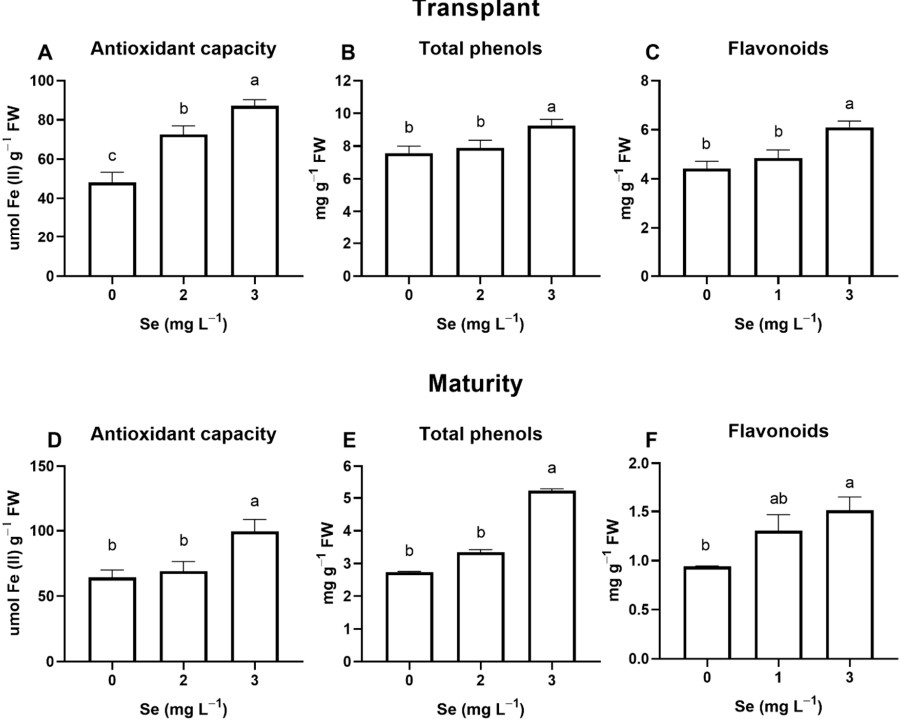

**Figure 9.** Antioxidant capacity, total phenol and flavonoid contents, measured at transplant (**A**–**C**) and maturity (**D**–**F**), in leaves of basil plants treated with different concentrations of Se at pre-transplanting. Means (*n* = 3) flanked by different letter are statistically different for *p* = 0.05 after Tukey's test.

### 3.2.2. Mature Plants

The treatments with Se had no significant effect on the production of fresh and dry biomass, or on the content of dry matter of basil plants at maturity (Figure 6D–F).

Increasing doses of Se added to the substrate resulted in an increasing leaf Se concentration of mature plants. The exposure to 3 mg Se $L^{-1}$ induced a leaf Se content which was 77.7% higher in comparison to the control plants. In general, the Se content in Se-fortified basil plants at maturity tended to be lower compared to seedlings (Figure 7B).

The contents of macroelements were not affected by the exposure to Se, except for the Mg content, which was 12.9% higher in plants treated with 3 mg Se $L^{-1}$, compared to the control (Table 4). Conversely, the Se treatments decreased the concentration of microelements. Both 2 and 3 mg Se $L^{-1}$ led to a lower concentration of Cu ($-47.7$% with 2 or $-52.3$ with 3 mg $L^{-1}$ of Se), Mn ($-47.7$% with 2 or $-52.3$% with 3 mg $L^{-1}$ of Se) and Zn ($-14.1$% with 2 or $-12.0$% with 3 mg $L^{-1}$ of Se). The concentration of Fe was only reduced ($-23.1$%) by the 3 mg Se $L^{-1}$ treatment, but not by the other treatments (Table 5). Treatments with Se did not affect the leaf nitrate content (Figure 10).

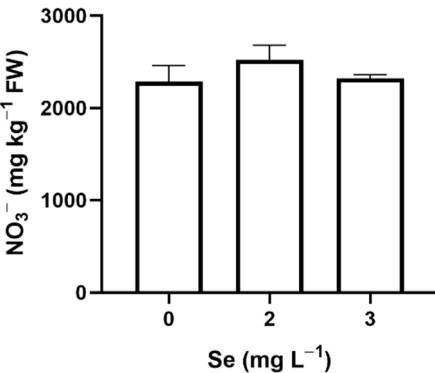

**Figure 10.** Nitrates ($NO_3^-$) content, measured at maturity in leaves of basil plants treated with different concentrations of Se at pre-transplanting. Means ($n = 3$) flanked by different letter are statistically different for $p = 0.05$ after Tukey's test.

The addition of Se to the substrate did not affect the total chlorophyll and carotenoid contents (Figure 8C,D), but the exposure to 3 mg Se $L^{-1}$ increased the leaf antioxidant capacity and total phenol and flavonoid contents (Figure 9D–F).

## 4. Discussion

### 4.1. Effect of Se Treatments on Plant Se Accumulation

The results confirmed that lettuce and basil seedlings are able to accumulate Se when selenate is supplied to the substrate at pre-transplanting. The ability of seedlings to take up Se from the growing substrate has been detected in broccoli seedlings exposed to 3.95 mg Se $L^{-1}$ by Bachiega et al. [39].

Funes-Collado et al. [40] found that the biofortification of vegetables through Se-enrichment of peat was effective in terms of the biofortification of the adult plants of cabbage, lettuce, chard and parsley without inducing negative effects on the biomass production at similar concentrations (2–8 mg Se $kg^{-1}$) to those used in our study. In our experiment the fresh biomass was lower only in basil seedlings exposed to 3 mg Se $L^{-1}$.

At maturity, after cultivation in the open field, plants biofortified with 3 mg Se $L^{-1}$ showed higher leaf Se contents compared to the other treatments. To the best of our knowledges, the only study on the biofortification of vegetables though substrate enrichment at pre-transplanting was conducted by Businelli et al. [23]. Despite the lower concentration used in our experiment, the leaf Se concentrations that we found in lettuce were higher than the concentration found by Businelli et al. [23] in lettuce plants grown from seedlings treated with 0, 10 and 20 mg Se $kg^{-1}$ of dry peat. The leaf Se concentration does not depend only on the dose of Se applied to the substrate, but also on environmental conditions and

cropping techniques. For instance, a deficient amount of water in soil strongly limits the root uptake of minerals even in fertilized soils. A significant lower Se content has been detected in lettuce plants during drought due to limited precipitation [41]. This means that a specific cultivation protocol is required for the production of biofortified seedlings.

In our experiment the Se content detected in mature plants of lettuce was lower compared to the results obtained in lettuce [17,18,23,25,42] and basil [12,22,30] plants, at the baby-leaf or maturity stage, and biofortified during the post-transplant period. This difference might be due to the total dose of Se applied, the duration of treatment, and the method of Se application.

In our experiment the maximum total amount of Se applied to each plant was 0.18 mg, whereas the supply of Se in previous post-transplant biofortification experiments ranged from 50 mg L$^{-1}$ in lettuce [18] to 600 mg L$^{-1}$ in basil [30]. In addition, in the previous experiments conducted by Puccinelli et al. [18,30], lettuce and basil plants were cultivated in a nutrient solution containing Se throughout the growing period.

At harvest, the leaf Se content in plants enriched with $Na_2SeO_4$ varied between 70 and 116 ug kg$^{-1}$ FW in lettuce, and between 61 and 87 ug kg$^{-1}$ FW in basil. In our study a serving size of 100 g of Se-biofortified leaves contained from 6.98 to 11.57 μg of Se in lettuce and from 6.07 to 8.71 μg of Se in basil. The consumption of 100 g of lettuce and basil would provide from 10% to 17% and from 9% to 12% of the adequate intake (AI) of Se [4], respectively.

*4.2. Effect of Se Treatments on Plant Growth*

In our study, dry biomass production was never affected by the Se treatments. Our results are in agreement with previous findings for basil [21], lettuce [18,43] and spinach [44] in which these plants were treated with similar concentration of Se. The toxic effects of Se in plants can be related to interference with S metabolism [5], which results in leaf chlorosis and a decrease in protein synthesis and dry matter production [45]. Se toxicity generally occurs at higher Se leaf concentrations than those detected in our experiment [46]. The reduction in fresh biomass at the transplanting of basil plants treated with 3 mg L$^{-1}$ might be due to a reduction in water content, and the dry biomass did not show any decrease.

*4.3. Effect of Se Treatments on Leaf Mineral Content*

Contrasting results have been reported in the literature regarding the effect of Se supplementation on plant mineral content. The influence of Se supply on macro and micronutrient concentration is not entirely clear, but it appears to differ in a genotype-specific way.

In our experiment, when plants were exposed to Se, macroelements tended to increase at transplanting, and microelements tended to increase at harvest in lettuce, whereas the opposite effect was found in basil. Our results are consistent with studies conducted on mature plants exposed to Se, in which the contents of K, P, Ca, and Mg increased or decreased depending on the species and/or cultivar [19,47,48]. In an experiment with Se-enriched microgreens, conducted by Pannico et al. [49], treatment with Se increased the content of P, K, Ca and Mg in basil and coriander, but decreased the content of the same elements in tatsoi. In the same work, contrasting results were obtained for Na content [49].

Conflicting results have been reported on micronutrient concentration in Se enriched plants. Pannico et al. [49] found an increased content of Fe, Zn and Mn in Se-enriched microgreens of basil and coriander, but a lower content in Se-enriched microgreens of tatsoi. Rios et al. [47] observed an increased level of iron and manganese in lettuce after Se applications, whereas Wu and Huang [48] reported a reduction in zinc and manganese in white clover treated with Se.

The maximum levels for nitrates in leafy vegetables are defined by EU regulation 1258/2011. These limits range between 2000 and 7000 mg kg$^{-1}$ FW depending on the plant species, growing cycle, and type of cultivation, and are higher for autumn-winter grown vegetables in comparison to spring-summer [50]. In our work, the nitrate content in lettuce and basil leaves was always lower than the maximum value allowed for lettuce

grown in spring-summer (3000 mg kg$^{-1}$ FW). The reduction in leaf nitrate content observed in lettuce leaves exposed to 3 mg Se L$^{-1}$ is consistent with previous results obtained in Se-enriched lettuce plants [19,47,51,52] and microgreens of coriander, basil and tatsoi [49]. The lower leaf nitrate content could be due to the promotion of nitrate reductase and glutamate synthase activities [17] or to the antagonistic effect of selenate exerted on nitrate anions [52]. In addition, in our experiment Se did not affect the leaf nitrate content in basil, at any of the stages, and in lettuce at transplant, which is in agreement with the experiment conducted by Puccinelli et al. [22] in lettuce and chicory.

### 4.4. Effect of Se Treatments on Leaf Quality Parameters

The decrease in chlorophyll content observed in seedlings exposed to 3 mg Se L$^{-1}$ may be due to the toxic effect of Se, and is in line with the results obtained in lettuce [53], and in spinach plants treated with Se concentrations higher than 1 mg L$^{-1}$ [54].

In our experiment, the highest dose of Se induced a higher antioxidant capacity in both species at both sampling times, consistent with results obtained by Rady et al. [55]. The increase in antioxidant capacity and flavonoid content might be due to the stress induced by Se, which acted as a chemical eustressor. In fact, eustress, such as nutritional stress, can activate physiological and molecular mechanisms and stimulate the strategic accumulation of bioactive compounds to adapt the plant to an environment with suboptimal characteristics [56]. According to our results, previous studies have shown an increased flavonoid content in basil plants exposed to 3–20 mg L$^{-1}$ [57] or 12 mg L$^{-1}$ [22].

In plant cells, phenols can form an antioxidant system, for example ascorbate which alleviates the oxidative stress and counteracts the reduction in biomass [58]. Se can have several beneficial effects in plants, for example, upregulating the antioxidant defence system [59].

## 5. Conclusions

The main goal of this study was the selenium biofortification of lettuce and basil in early development stages. This new approach to produce vegetables with an appropriate Se content can increase dietary Se intake. Our biofortification technique produced Se-enriched basil and lettuce seedlings for transplant with minimal or no changes to conventional cultivation techniques. A small quantity of Se at pre-transplanting reduced the amount of Se released into the environment. Furthermore, the addition of sodium selenate to the water used to wet the substrate immediately after sowing only represents a minor change in the seedling production process. With our technique, biofortified seedlings and mature plants of basil and lettuce can be obtained.

More research is needed to evaluate the application of higher concentration of Se for the pre-transplant biofortification of basil and lettuce. Additional investigations should be conducted to study the application of this technique to produce other Se-biofortified leafy species or fruity vegetables.

**Author Contributions:** Conceptualization, M.P., F.M. and B.P.; data curation, M.P.; formal analysis, M.P., L.P. and I.R.; investigation, M.P., F.M. and B.P.; methodology, M.P., F.M. and B.P.; software, M.P.; supervision, M.P., F.M. and B.P.; validation, M.P., F.M. and B.P.; writing—original draft, M.P. and B.P.; writing—review & editing, M.P., F.M. and B.P. All authors have read and agreed to the published version of the manuscript.

**Funding:** This research received no external funding.

**Institutional Review Board Statement:** Not applicable.

**Informed Consent Statement:** Not applicable.

**Data Availability Statement:** Not applicable.

**Acknowledgments:** The authors would like to thank "L'Ortofruttifero" for their support in providing the seeds and producing the seedlings.

**Conflicts of Interest:** The authors declare no conflict of interest.

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
