# Peer review of "Biofortification of Lettuce and Basil Seedlings to Produce Selenium Enriched Leafy Vegetables"

_horticulturae, doi:10.3390/horticulturae8090801_

Round 1

Reviewer 1 Report

Review of the article: “ Biofortification of lettuce and basil seedlings to produce sele-2 nium enriched leafy vegetables  

 The authors investigated how selenium fertilization at the plant seedling production stage would affect the growth and development of lettuce and basil seedlings. In addition, the aim of the research was to determine whether such a seedling production technology would consequently achieve the effect of enriching these vegetables with selenium during cultivation in the field.

Therefore, the authors investigated a new procedure / protocol for enriching seedlings and mature plants (at maturity phase) with selenium

 This manuscript have a very good synthetic introduction - literature review. Only in the third paragraph (P.1, L39-42) please specify that selenium (VI) (SeO4) interacts with sulfur (S-SO4). In contrast, selenium (IV) (SeO3) has an interaction with phosphorus. Cite relevant literature on this topic.

Material and methods

Move the heading of table 1 (P.2 L 97) to page no.3.

 I cannot understand why phosphorus was not determined by ICP-MS like Na, K, Ca, Mg, Cu, Fe, Mn, and Zn. However, the test technique used (UV / VIS spectrometry) is acceptable.

 2.4. Statistical Analysis Data . Statistical analysis was performed us-152 ing R Statistical Software – please complete this sentence with the name of the manufacturer of this software and enter the address / city / country.

 3. Results

 Basically, despite the separation of sub-chapters for lettuce and basil, please use the words "lettuce plants" or "basil plants” - in the description of the results .

Table 3. Please verify that there are significant differences for Na content for the Maturity phase. There are n.s in the table but there are the same homogeneous groups "C". Consequently, correct the description of the results and the discussion for "Na" in Table 3 accordingly.

 P.6. L 190-192. Please verify the description for "flavonoid". It should probably be "+ 6.10%" and not "- 6.10%."

 P.9,10 and 11. Figures 2, 3 and 4 are incorrectly numbered and should be Figure 5, 6 and 7. Therefore please check and correct the cross-references to the relevant figures throughout the manuscript accordingly.

 4. Discussion

 P.12 L 302. Error in the way of citing “Businelli et al. (2015). Should there be Businelli et al. [22]).

P. 13 L Error in the way of citing Rady et al., (2020). Moreover, on page 13, lines 377-386, there is an error in the continuity of the numbering of the cited literature - wrong citation order for items 53-56.

 Complete the discussions (Chapter 4.1) about the potential influence of climatic conditions on the accumulation of selenium in plants. Please consider citing the publication: "Iodine and selenium biofortification of lettuce (Lactuca sativa L.) by soil fertilization with various compounds of these elements." Acta Scientiarum Polonorum Hortorum Cultus, 15 (5): 69-91. Alternatively, any other, which describes the influence of climatic conditions on biofortification of vegetables into selenium in field cultivation

Author Response

Only in the third paragraph (P.1, L39-42) please specify that selenium (VI) (SeO4) interacts with sulfur (S-SO4). In contrast, selenium (IV) (SeO3) has an interaction with phosphorus. Cite relevant literature on this topic.

Text has been corrected

Material and methods

Move the heading of table 1 (P.2 L 97) to page no.3.

Done

 I cannot understand why phosphorus was not determined by ICP-MS like Na, K, Ca, Mg, Cu, Fe, Mn, and Zn. However, the test technique used (UV / VIS spectrometry) is acceptable.

 2.4. Statistical Analysis Data . Statistical analysis was performed us-152 ing R Statistical Software – please complete this sentence with the name of the manufacturer of this software and enter the address / city / country.

The information has been added

  1. Results

Basically, despite the separation of sub-chapters for lettuce and basil, please use the words "lettuce plants" or "basil plants” - in the description of the results .

Text has been corrected

Table 3. Please verify that there are significant differences for Na content for the Maturity phase. There are n.s in the table but there are the same homogeneous groups "C". Consequently, correct the description of the results and the discussion for "Na" in Table 3 accordingly.

The table has been corrected.

P.6. L 190-192. Please verify the description for "flavonoid". It should probably be "+ 6.10%" and not "- 6.10%."

Text has been corrected

P.9,10 and 11. Figures 2, 3 and 4 are incorrectly numbered and should be Figure 5, 6 and 7. Therefore please check and correct the cross-references to the relevant figures throughout the manuscript accordingly.

The number of the figures and the references in the text have been corrected

 Discussion

P.12 L 302. Error in the way of citing “Businelli et al. (2015). Should there be Businelli et al. [22]).

Text has been corrected

  1. 13 L Error in the way of citing Rady et al., (2020).

Text has been corrected

Moreover, on page 13, lines 377-386, there is an error in the continuity of the numbering of the cited literature - wrong citation order for items 53-56

The citation order has been corrected

 Complete the discussions (Chapter 4.1) about the potential influence of climatic conditions on the accumulation of selenium in plants. Please consider citing the publication: "Iodine and selenium biofortification of lettuce (Lactuca sativa L.) by soil fertilization with various compounds of these elements." Acta Scientiarum Polonorum Hortorum Cultus, 15 (5): 69-91. Alternatively, any other, which describes the influence of climatic conditions on biofortification of vegetables into selenium in field cultivation

The information and the citation have been added.

Reviewer 2 Report

Line 87-89 Authors present available forms mcroelements ( P <lq, available Ca 1.45 g kg-1, available Mg 0.12 g 87 kg-1, available Na 0.16 g kg-1, available K 0.24 g kg-1, total C 0.12%, organic C 1.23%, N 1.2 88%), but should give some information what method use to determine content of this elements. Available form of P we can use few method (Olsen method or Egner-Riehm method or Melih 3 or…..)

Line 112-113 - Authors should explain why they dried plants at 50° C? The dry weight is usually determined at 105° C.

Line 252 Improve (P-PO4-),

Authors should improve goal of this manuscript

The main goal of the study is the biofortification of plants (lettuce and basil) into selenium in the early development stages. This is a new approach to biofortification of plants into selenium.

Author Response

Line 87-89 Authors present available forms mcroelements ( P <lq, available Ca 1.45 g kg-1, available Mg 0.12 g 87 kg-1, available Na 0.16 g kg-1, available K 0.24 g kg-1, total C 0.12%, organic C 1.23%, N 1.2 88%), but should give some information what method use to determine content of this elements. Available form of P we can use few method (Olsen method or Egner-Riehm method or Melih 3 or…..)

The physical and chemical properties of the soil have been determined according to the methods described in “Methods of soil analysis, part 1 and 3 Physical and chemical methods. ASA-SSSA, Madison.”
The reference has been added to the text.

Line 112-113 - Authors should explain why they dried plants at 50° C? The dry weight is usually determined at 105° C.

This information has been added to the text

Line 252 Improve (P-PO4-),

Text has been corrected

Authors should improve goal of this manuscript.

The main goal of the study is the biofortification of plants (lettuce and basil) into selenium in the early development stages. This is a new approach to biofortification of plants into selenium.

Text has been improved (see Conclusions)

Reviewer 3 Report

Reviewer:

The manuscript is entitled " Biofortification of lettuce and basil seedlings to produce selenium enriched leafy vegetables." has a good research idea. In this research, selenium treatment was carried out at the seedling stage of vegetables, and selenium treatment was no longer carried out after transplanting, which was simpler and more efficient. The original vegetable production process has not been changed basically, and the yield after maturity has not decreased significantly, which is convenient for popularization and application. As an edible vegetable, the form of selenium is an important index for the quality evaluation of selenium-rich vegetables. It is suggested that the authors supplement the determination data of selenium form in edible parts and analyze the relative changes of selenium form content before and after transplant. Specific comments are as follows:

Line 154: Please add photos of plants before and after treatment.

Lines 161-186: Please change Table 2, Table 3, and Table 4 to Fig.

Line 178: Supplementary data on selenium speciation in edible parts.

Author Response

The manuscript is entitled " Biofortification of lettuce and basil seedlings to produce selenium enriched leafy vegetables." has a good research idea. In this research, selenium treatment was carried out at the seedling stage of vegetables, and selenium treatment was no longer carried out after transplanting, which was simpler and more efficient. The original vegetable production process has not been changed basically, and the yield after maturity has not decreased significantly, which is convenient for popularization and application. As an edible vegetable, the form of selenium is an important index for the quality evaluation of selenium-rich vegetables.

It is suggested that the authors supplement the determination data of selenium form in edible parts and analyze the relative changes of selenium form content before and after transplant.

This is a preliminary work focused on the application of an innovative technique for the biofortification of leafy vegetables. Since the reference value for adequate and toxic amount of Se, proposed by EFSA, are referred to the total Se content, we evaluated the percentage of the satisfaction of the daily adequate intake and not the direct effects of Se on human health. However, we found the suggestion of the reviewer very interesting, and we will take account of it for future studies.

Specific comments are as follows:

Line 154: Please add photos of plants before and after treatment.

The treatments were performed immediately after sowing when seedlings had not yet emerged

Lines 161-186Please change Table 2, Table 3, and Table 4 to Fig.

We changed Tables 2 and 5 to figures. We did not change Tables 3 and 4 to figures in order to have a similar number of figures and tables.

Line 178: Supplementary data on selenium speciation in edible parts.

See the above answer to the first comment.

Reviewer 4 Report

This manuscript titled ‘’Biofortification of lettuce and basil seedlings to produce selenium enriched leafy vegetables” evaluated the effects of different dosage of Se on the growth, Se content, mineral element content, and antioxidant indexes in lettuce and sweet basil at two growth stages. This manuscript fits the scope of the journal. But the manuscript must be revised before considering for acceptance. My concerns are as follows.

Line 31 EFSA should be in full, not abbreviation here.

Introduction section: too many paragraphs. Some of them can be merged. E.g. paragraph 2 and 3.

Line 43-47 The sentences should be reorganized because some of the information are replicated.

Line 87 available P <lqIt is confusing.

Line 115 --2.3.2 Selenium content: A brief description of the determination should be added because the not all the readers know the method. Refer to 2.3.3 Mineral content: the method is clear.

Line 152 p < 0.05, P value should be italics in capital letters: P < 0.05. Also, this issue needs to be corrected in the table footnote or figure captions.

Line 166, 205, 208, selenium—Se, also check this case throughout the manuscript.

Line 231: Figure 2? is it Figure 5? Overall, the figure order is chaotic. I find two Figure 2 and two Figure 3 in the manuscript. The authors should be careful to address this issue.

Line 297 Discussion in this paragraph is good.

Line 390 remove “and disease”, “a higher Se content”—“an appropriate Se content”, because more Se is not always better for crops or humans.

Reference format is not consistent. References must be corrected following the journal’s requirements. 

Author Response

This manuscript titled ‘’Biofortification of lettuce and basil seedlings to produce selenium enriched leafy vegetables” evaluated the effects of different dosage of Se on the growth, Se content, mineral element content, and antioxidant indexes in lettuce and sweet basil at two growth stages. This manuscript fits the scope of the journal. But the manuscript must be revised before considering for acceptance. My concerns are as follows.

Line 31 EFSA should be in full, not abbreviation here.

Text has been corrected.

Introduction section: too many paragraphs. Some of them can be merged. E.g. paragraph 2 and 3.

Paragraphs 2, 3 and 4 have been merged.

Paragraphs 5, 6, 7 and 8 have been merged.

Line 43-47 The sentences should be reorganized because some of the information are replicated.

The sentences have been modified, in order to make it clear that some information refer to results obtained in experiment conducted on basil plants.

 Line 87 available P <lq?It is confusing.

Text has been corrected.

Line 115 --2.3.2 Selenium content: A brief description of the determination should be added because the not all the readers know the method. Refer to 2.3.3 Mineral content: the method is clear.

A brief description of the method has been added.

Line 152 p < 0.05, P value should be italics in capital letters: P < 0.05. Also, this issue needs to be corrected in the table footnote or figure captions.

Text has been corrected.

Line 166, 205, 208, selenium—Se, also check this case throughout the manuscript.

Text has been corrected.

 Line 231: Figure 2? is it Figure 5? Overall, the figure order is chaotic. I find two Figure 2 and two Figure 3 in the manuscript. The authors should be careful to address this issue.

The number of the figures and the references in the text have been corrected.

Line 297 Discussion in this paragraph is good.

Line 390 remove “and disease”, “a higher Se content”—“an appropriate Se content”, because more Se is not always better for crops or humans.

Text has been corrected.

Reference format is not consistent. References must be corrected following the journal’s requirements. 

Reference format has been checked and corrected according to the Journal guidelines.

Round 2

Reviewer 3 Report

Through experiments basically confirmed that this formula can increase the total selenium content of vegetables, while reducing the use of selenium and environmental pollution. However, I think it is important to study the content of organic selenium in vegetables to enhance food safety. It is recommended that subsequent studies consider increasing this measurement. Finally, it is asked whether the peat attached to the roots is tried to be removed during transplanting, which may bring additional errors and pollution.

Line 78:Whether the root system contains selenium peat when transplanting.

Author Response

Through experiments basically confirmed that this formula can increase the total selenium content of vegetables, while reducing the use of selenium and environmental pollution. However, I think it is important to study the content of organic selenium in vegetables to enhance food safety. It is recommended that subsequent studies consider increasing this measurement.

Thank you again for your suggestion. As already stated in the Answers to the previous Review Report (Round 1), we will take account of it for future studies.

Finally, it is asked whether the peat attached to the roots is tried to be removed during transplanting, which may bring additional errors and pollution.

Line 78:Whether the root system contains selenium peat when transplanting.

The peat attached to the roots was not removed. Since the aim of the study was to improve the selenium uptake by plants, it would not be not worth removing the peat and the selenium accumulated in it.